# RNA-Binding Proteins in Bladder Cancer

**DOI:** 10.3390/cancers15041150

**Published:** 2023-02-10

**Authors:** Yuanhui Gao, Hui Cao, Denggao Huang, Linlin Zheng, Zhenyu Nie, Shufang Zhang

**Affiliations:** Central Laboratory, Affiliated Haikou Hospital of Xiangya Medical College, Central South University, Haikou 570208, China

**Keywords:** RNA binding proteins, bladder cancer, LIN28B, human antigen R, heterogeneous nuclear RNPs

## Abstract

**Simple Summary:**

Bladder cancer (BC) is a common malignant tumor of the urinary system. Despite extensive advances in the treatment of BC, it remains one of the most recurring and life-threatening tumors. At present, there have been increasing reports of studies on the presence of aberrant regulation of RBPs in BC. However, the complex regulatory network of these RBPs in BC remains to be fully elaborated. RBPs have a very high potential in tumor prediction and personalized therapy. Moreover, only with a deep understanding of their regulatory mechanisms, expression characteristics, and potential binding sites, among other issues, will it become possible to apply RBPs to clinical applications. This article aims to summarize the research progress of RBPs in BC. It also attempts to encourage clinicians and researchers to devote attention this field of study and provides a reference for researchers who aspire to pursue a career in this area.

**Abstract:**

RNA-binding proteins (RBPs) are key regulators of transcription and translation, with highly dynamic spatio-temporal regulation. They are usually involved in the regulation of RNA splicing, polyadenylation, and mRNA stability and mediate processes such as mRNA localization and translation, thereby affecting the RNA life cycle and causing the production of abnormal protein phenotypes that lead to tumorigenesis and development. Accumulating evidence supports that RBPs play critical roles in vital life processes, such as bladder cancer initiation, progression, metastasis, and drug resistance. Uncovering the regulatory mechanisms of RBPs in bladder cancer is aimed at addressing the occurrence and progression of bladder cancer and finding new therapies for cancer treatment. This article reviews the effects and mechanisms of several RBPs on bladder cancer and summarizes the different types of RBPs involved in the progression of bladder cancer and the potential molecular mechanisms by which they are regulated, with a view to providing information for basic and clinical researchers.

## 1. Introduction

Tumor formation in humans is an extremely complex and multi-stage process that typically occurs over years or decades. The histiocytes of normal individuals gradually develop into tumors with malignant phenotypes through evolution, through a process called tumor progression. Tumors can occur in various tissues of the human body, and the incidence increases with age. Only in very rare cases do cancerous cells progress to clinically visible tumor tissue with occupying lesions. Tumor progression is closely related to epigenetics, RNA post-transcriptional modification, protein post-translational modification, and other life processes. Tumors are not only regulated by these life processes but are also affected by normal biochemical reactions, reshaping cellular life activities and ultimately giving cells the ability to proliferate indefinitely.

Bladder cancer is the commonest malignant tumor in the urinary system. According to cancer statistics released in 2023, the estimated incidence of bladder cancer in the United States is 82,290 cases, and the mortality rate is 16,170 cases [1]. The incidence is higher than that in 2022, while the mortality is slightly lower than that in 2022 [2]. According to the report published by China, the incidence of bladder cancer in 2020 was 91,893 cases, with mortality in 42,973 cases [3]. Bladder cancer is a broad concept that encompasses everything from low-risk non-muscle-invasive bladder cancer to high-risk primary invasive bladder cancer. Low- and intermediate-risk non-muscle invasive bladder cancer (NMIBC) patients face high recurrence rates, with 5-year event-free survival rates reaching 43% and 33% [4]. Metastasis of bladder cancer is a catastrophe that 50–70% of muscle invasive bladder cancer (MIBC) patients have to face, and given the extremely high metastasis rate, the 5-year overall survival rate for advanced MIBC is 4.8% [5]. The research on bladder cancer not only requires the development of new biomarkers for diagnosis and molecular targets for personalized therapy but also in-depth studies of its progression, recurrence, metastasis, and other processes.

RBPs play crucial roles in the regulation of cellular life processes, especially RNA splicing, modification, transport, localization, stabilization, degradation, and translation. Certain RBPs are expressed in a variety of cells to maintain essential cellular functions. Altered structure or disturbed expression of RBPs may cause different diseases, and this concept is reflected in tumorigenesis [6]. Given that RBPs can regulate post-transcriptional RNA, they can rapidly and efficiently alter gene expression in response to changes in the microenvironment. A single RBP can bind multiple targets, and different combinations of several RNP interactions contribute to enhanced cellular recognition and responses to stress [7]. In addition, RBP can promote mRNA translation by recruiting specific translation signaling molecules [8]. By contrast, RBPs involved in the RNA-induced silencing complex can inhibit target mRNA translation while inducing its degradation [9,10]. In several cases, two RBPs can compete in binding the same segment of mRNA, e.g., CUGBP2 and human antigen R (HuR) can bind to COX-2 mRNA, and in HCT-116 cell lines treated with radiotherapy, the RBP bound to COX-2 mRNA shifts mainly from HuR to CUGBP2 and inhibits the translation of COX-2 mRNA [11]. Certain RBPs play contrasting roles in a variety of different tumor cells, such as insulin-like growth factor 2 mRNA binding protein 1, also known as IMP1, which promotes colorectal cancer [12] and liver cancer [13] occurrence, progression, and metastasis; however, it can inhibit the proliferation and metastasis of breast cancer [14]. Notably, a high stromal cell IMP1 expression in the colon cancer tumor microenvironment suppresses tumorigenesis, whereas the deletion of stromal IMP1 forms a microenvironment that promotes colon carcinogenesis [15]. However, several RBPs are only expressed in specific tumor cells; PAT1 homolog 2 is highly expressed in renal chromophobe cancer, whereas dihydrouridine synthase 1 like is barely highly expressed in bladder cancer [6].

Given that RBPs control gene expression mainly at the post-transcriptional level, the RRM domain can participate in blocking certain sites by binding RNA through a canonical RRM domain in its cap-binding complex [16,17]. The RRM structural domain of RBP binds to RNA and alters the secondary structure of RNA; thus, RBP influences the entry of mRNA initiation factors into the ribosomal subunit [18], which ultimately regulates the activity of certain kinases in tumors. In addition, RRM plays a crucial role in precursor mRNA (pre-mRNA) splicing [19].

RBPs play an important role in biochemical processes, such as tumorigenesis, progression, invasion, metastasis, and drug resistance. In particular, RBPs are involved in the development, progression, and metastasis of bladder cancer [20], as well as in predicting survival of bladder cancer patients [21]; however, the role of RBPs in the development and progression of bladder cancer is unclear. We hereby summarize and review several RBPs that play a major role in the development and progression of bladder cancer. This review aims to provide a detailed characterization of the RBPs associated with bladder cancer and focuses on their structure (especially RRM), function, interactions, causative pathogenic effects, and resulting treatment and prognosis. It also attempts to encourage clinicians and researchers to devote themselves to this field of study and provides a reference for researchers who aspire to pursue a career in this area.

## 2. RNA Binding Motif 3 (RBM3)

RBM3 is a glycine-rich cold shock protein whose expression can be stimulated by hypothermia, ischemia, or hypoxia [22,23,24]. RBM3 has two highly conserved RRMs, namely RNP1 and RNP2, at the N terminus and an arginine-glycine-rich domain (RGG) at the C terminus [16]. The RGG structural domain mainly regulates the process of RNA cleavage and polyadenylate cyclization [25]. The RGG structural domain, especially the part with arginine residues, is essential for mRNA export, because the absence of a single arginine residue in the RGG structural domain can interrupt the shuttling process of RBM3 between the nucleus and cytoplasm [25,26]. RBM3 performs four main functions in tumors.

RBM3 can bind and affect the translation of mRNA. It influences mRNA stability and the translation of COX-2, interleukin (IL)-8, and vascular endothelial growth factor (VEGF) [27]. In general, RBM3 facilitates the translation of various mRNAs into proteins [26,27,28]. This promotion includes the following mechanisms: (1) binding to the 60S ribosomal subunit in an RNA-independent manner [28]; (2) increased formation of active polyribosomes [27]; (3) dephosphorylation of eukaryotic initiation factor (eIF2α); (4) promotion of eIF4E phosphorylation [28].

Under low-temperature conditions, RBM3 can alter miRNA levels and thus promote protein translation. RBM3 binds to a precursor miRNA and facilitates its processing by the Dicer complex to form a mature double-stranded miRNA [29]. The regulation of miRNAs by RBM3 is two-sided: it can positively regulate most miRNAs, but reducing the level of RBM3 can promote the expression of a small number of temperature-sensitive miRNAs, thereby preventing pathological hyperthermia [30]. These results suggest that RBM3 is essential for the mitotic process of cells.

RBM3 can play a regulatory role in the cell cycle of G2/M transition. In colorectal cancer cells, RBM3 induces stem cell proliferation through a mechanism that increases β-catenin signaling by inhibiting glycogen synthase kinase-3 beta kinase activity [31]. By contrast, knockdown of RBM3 expression in the human HCT116 colon cancer cell line caused increases in caspase-dependent apoptosis, nuclear cyclin B1 expression, and Cdc25c, Chk1, and Chk2 phosphorylation levels, which are a series of alterations suggesting that downregulation of RBM3 will prevent cell mitosis [27]. In vivo, embryonic fibroblasts from RBM3-deficient mice showed a significant increase in the number of cells in the G2 phase [32]. These conclusions can also explain the higher sensitivity of tumors with a high RBM3 expression to chemotherapy than those with low or negative expression [33].

When cells receive various external stimuli, unfolded proteins accumulate in the lumen of the endoplasmic reticulum and activate the unfolded protein response (UPR) to rescue cells. Sustained and/or intense endoplasmic reticulum stress (ERS) induces apoptosis [34]. The protein kinase R-like endoplasmic reticulum kinase (PERK)-eIF2α-C/EBP-homologous protein (CHOP) signaling pathway plays an important role in UPR-induced apoptosis [35]. Under sustained ERS, RBM3 can inhibit the phosphorylation of PERK and eIF2α, causing a decrease in the expression of CHOP and inhibiting UPR to avoid apoptosis [36]. This condition may be the reason why the UPR does not induce apoptosis despite its low-temperature activation; low-temperature-induced RBM3 may play an important role in this process [37]. Hypothermia can also alleviate ischemia-induced apoptosis by inhibiting the UPR [38].

In bladder cancer, the role of RBM3 is similar to that in other tumors. A clinical retrospective study including 259 bladder cancer patients showed that the low expression of RBM3 was an independent factor for poor prognosis of bladder cancer [39]; this finding is closely related to the progression of bladder cancer and decreased overall survival of patients [40]. A similar study revealed that patients with a high expression of RBM3 were associated not only with a low tumor grade but also with a low risk of lymphovascular invasion (lymph node invasion) [41]. The effect of RBM3 on bladder cancer may depend on several factors: (1) the expression level of RBM3 is closely related to tumor stage; (2) RBM3 silencing can increase the number of G2/M stage cells and eventually lead to apoptosis [33]; (3) RBM3 directly binds to a variety of mRNAs, thus regulating the activity of multiple kinases in tumors [42,43]. For this stage, studies of RBM3 and bladder cancer have relied on the analysis of clinical samples in immunohistochemistry, but more basic mechanistic studies are lacking. In particular, mechanistic studies on how RBM3 limits the development and progression of bladder cancer are limited. Clarifying these issues will not only further elucidate the mechanisms of bladder cancer development but also improve the current status of bladder cancer treatment and provide personalized therapeutic targets. In addition, the function of RBM3 in tumors and the mechanistic pathways it depends on can be elucidated and extended to more tumor treatments.

## 3. LIN28

LIN28 is a highly conserved RNA-binding protein in eukaryotes [44]. In a variety of mammals, including humans, LIN28 is divided into LIN28A and its homologous molecule, LIN28B [45]. Human LIN28A is encoded by the Lin28a gene, which is located on chromosome 1p36.11 and is mainly expressed in embryonic stem cells and embryonic carcinoma cells [46,47]. LIN28B is encoded by the Lin28b gene on chromosome 6q21 and is mainly expressed in the testis, placenta, and other tissues [45,48]. LIN28A and LIN28B have highly similar protein structures: both have two functional domains, namely a cold shock protein domain (CSD) and a retroviral zinc finger (cys-cys-his-cys, CCHC) domain [46]. After mutation of either domain, the other domain still has the function of binding RNA, suggesting that CSD and CCHC can participate in the RNA binding of LIN28 [49]. LIN28 protein is mainly localized in the cytoplasm [47]. However, it can also be present in RNPs, polyribosomes (polysome), P vesicles, and stress granules [50]. Meanwhile, LIN28B is mainly located in the nucleus, and it may exert its biological function through the cytoplasmic microprocessor [51]. However, the expressions of LIN28A and LIN28B are mutually exclusive, and tumor cells expressing LIN28A do not express LIN28B, and vice versa [51].

The miRNA let-7 family contains 12 miRNA members, which act as tumor suppressors and inhibit the expression of a variety of important oncogenes (including Ras, Myc, and so on) by binding to their 3′ untranslated regions [52,53,54]. This function of let-7 is regulated by the RNA-binding protein LIN28 [55]. Overexpression of LIN28A or LIN28B is associated with a variety of tumors, leading to increased tumor aggressiveness and poorer prognosis [56]. LIN28B has also attracted considerable attention as one of the downstream genes of nuclear factor (NF)-κB [57]. LIN28A and LIN28B can inhibit the expression of oncogenes, such as Ras and Myc, by inhibiting let-7-miRNA [58,59]. MiRNA let-7 acts as the main effector molecule of LIN28A and LIN28B, with which they form multiple feedback loops: (1) LIN28A/B can inhibit the maturation of let-7 through various mechanisms, whereas let-7 can inhibit the translation of LIN28A/B at the post-transcriptional level, reducing the protein expression level [60]; (2) after LIN28A/B inhibits the maturation of let-7, the inhibition of c-Myc by let-7 is relieved, and c-Myc can promote the transcription of LIN28A/B, forming a positive feedback loop [61,62]; (3) NF-κB can induce the expression of LIN28B, and LIN28B inhibits the maturation of let-7, thus releasing the inhibitory effect of let-7 on the expression of IL-6, which can activate the expression of NF-κB, forming a positive feedback loop. Thus, linking inflammation and tumor further reveals the role played by inflammatory factors in the malignant transformation of cells (Figure 1) [63].

In NMIBC, the inhibition of LIN28 significantly increases the expression level of let-7 and simultaneously inhibits the cell viability and migration ability of NMIBC [64]. Moreover, the expression of LIN28 in high-grade bladder transitional carcinoma (TCC) is significantly higher than that in normal bladder tissue and low-grade TCC; LIN28 can promote the progression and differentiation of NMIBC through the Lin28/let-7/c-Myc pathway [65]. LIN28B can also promote the expression of MYC through the LIN28B/let-7a pathway and promote the proliferation, invasion, and metastasis of bladder cancer cells [66]. LIN28A inhibits the expression of lysosome-associated membrane glycoprotein 1 in bladder cancer cells, thereby promoting bladder cancer proliferation, migration, and invasion [67]. In addition, LIN28A/LIN28B can promote bladder cancer proliferation, invasion, and metastasis by activating the transforming growth factor-β/Smad signaling pathway to drive epithelial–mesenchymal transition (EMT) [68]. LIN28B is also one of the downstream targets of macroH2A1, a histone variant; knocking down the expression of macroH2A1 can increase the expression of LIN28B. Knocking down the expression of macroH2A1 resulted in increased expression of LIN28B and enhanced tumorigenicity, radioresistance, degeneration of reactive oxygen species (ROS), and increased sphere formation ability of bladder cancer cells [69].

In conclusion, LIN28A and its paralog LIN28B are RBPs closely associated with tumors, and they exert their biological function by inhibiting the biosynthesis of members of the tumor suppressor gene let-7 miRNA family or changing the translation efficiency of the mRNA they bind. In bladder cancer, LIN28AB can promote tumor proliferation, metastasis, and invasion through a let-7-dependent mechanism and is resistant to radiotherapy and ROS. In other tumors, tumor cell lines highly expressing LIN28 exhibited resistance to chemotherapeutic drugs [70]. LIN28AB is a potential tumor therapy target. Inhibition of LIN28AB induced the regression of xenografted human tumors in mice [56]; however, future research needs to determine how LIN28AB/let-7 is precisely regulated.

## 4. HuR

HuR is an embryonic lethal abnormal vision gene that includes four family members: HuB, HuC, HuD, and HuR. The first three are expressed mainly in neural tissues and germ cells and are associated with neurodevelopment, whereas HuR is commonly expressed in all human cells [71]. The human HuR gene (hHuR) is located on chromosome 19p13.2, which is closely related to chromosomal translocations and tumor carcinogenesis in human tumors [72]. HuR contains three RRMs and a hinge region in which RRM1 and RRM2 bind to adenine- and uracil-rich elements (AU-rich elements, AREs) in the target mRNA. By contrast, RRM3 can bind to the polyadenylate tail of the target mRNA. In normal conditions, AREs can accelerate the poly-A tail of mRNA to undergo deadenylation to destabilize mRNA [73]. Therefore, when HuR protein binds to these AREs, it can inhibit its own deadenylation and help mRNA to be protected from nuclease degradation during mRNA transport from nucleus to cytoplasm, thereby increasing mRNA stability and promoting mRNA translation; thus, HuR plays a role in post-transcriptional regulation [73,74]. The hinge region between RRM2 and RRM3 contains a 52-amino-acid HuR nucleoplasmic shuttling sequence, which is the main motif for post-translational modification of HuR and a key region for nucleoplasmic transport (Figure 2) [75,76].

The nucleoplasmic transport function is essential for HuR to exercise its biological functions. Under physiological conditions, HuR is mainly distributed in the nucleus. When cells are damaged by radiation [77], depleted of nutrients or energy substances [78,79], including by heat shock [80], viral infection [81], and chemotherapeutic drugs [82], or stimulated by cytokines [83], HuR shuttles from the nucleus to the cytoplasm. After activation through phosphorylation [84], methylation [85], and acetylation [86], HuR can shuttle from the nucleus to the cytoplasm, maintain the stability of bound mRNA, and promote mRNA translation [87,88]. Given that HuR can interact with a variety of cytokines that promote tumor progression (such as survivin [89], COX-2 [90], VEGF [91], low-density lipoprotein receptor-related protein [92], etc.), mRNAs stably bind and promote its translation, which in turn leads to the abnormal distribution of tumor-promoting factors in the nucleocytoplasm of tumor cells. Therefore, inhibition of the nucleocytoplasmic shuttling of HuR may also become a potential tumor therapy.

In bladder cancer tissues, the nuclearly expressed HuR is not significantly correlated with any pathological features. However, the highly expressed HuR in the cytoplasm plays an important role in the proliferation, progression, and survival of bladder cancer cells, and its expression is related to angiogenesis and clinical staging and grading [93,94]; in MIBC cells, the level of cytoplasmic HuR is significantly higher than that in NMIBC cells [95]. By contrast, the inhibition of HuR accumulation in the cytoplasm significantly increases the cytotoxicity of chemotherapeutic drugs, such as cisplatin and adriamycin, and inhibits the growth of xenografts in mouse bladder tumors [96]. In bladder cancer, HuR overexpression can promote bladder cancer proliferation, invasion, metastasis, and EMT by increasing the stability of polypyrimidine tract-binding protein 1 (PTBP1) to upregulate its expression [97], whereas destabilization of HuR causes a decrease in PTBP1 and inhibits bladder cancer progression [98]. In bladder cancer, HuR is a long non-coding whose other target RNA is thought to be HOX transcript antisense RNA (HOTAIR). HuR can stabilize HOTAIR mRNA, promote bladder cancer proliferation, migration, and invasion, and inhibit apoptosis. Overexpression of HOTAIR can also increase the expression of HuR and promote its accumulation in the cytoplasm, thus enhancing HOTAIR expression and forming a positive HuR-HOTAIR feedback loop [99]. The role of HuR in the development of bladder cancer is largely dependent on the function of the mRNA it binds. The complex and extensive regulatory molecules in tumors endow HuR with a double-edged sword effect. HuR also enhances the stability and promotes the translation of ubiquitin-specific protease 8 by binding to its mRNA, which subsequently promotes the ubiquitination and degradation of SOX2 by acting as an E3 ligase, thereby inhibiting the invasive ability of bladder cancer [100]. HuR also upregulates miR-494 expression by stabilizing and promoting JunB mRNA translation. Up-regulated miR-494 destabilizes c-Myc mRNA and inhibits its translation, ultimately suppressing c-Myc-dependent matrix metalloproteinase (MMP)-2 expression, i.e., inhibiting bladder cancer proliferation and invasion via the HuR/JunB/miR-494/c-Myc/MMP-2 axis [101].

The double-edged sword-like regulation mechanism of HuR on bladder cancer confirms the importance of HuR in bladder cancer. On the one hand, the activity of HuR is determined by its bound target mRNA, and on the other hand, it depends on the subcellular localization of HuR. More research is being focused on HuR as a target for tumor prevention and treatment. Nanocarriers [102] or cholesterol lipid nanocarriers [103] are used to transport siRNA to the tumor site, which allows it to bind to HuR mRNA and inhibit its translation, thereby inhibiting tumor growth. Inhibitors of HuR, such as MS-444 [104] or ChlA-F [100,101], have also been tested for the treatment of tumors. Regardless of the type of targeted therapy strategy, further research is needed on its potential on- and off-target effects. However, at present, significant inhibitory effects have been achieved regarding the malignant phenotype of tumor cells caused by the abnormal expression of HuR. We look forward to more therapeutic strategies targeting HuR in bladder cancer.

## 5. Heterogeneous Nuclear RNPs (hnRNPs)

HnRNPs act as RBPs by binding to pre-mRNA to form the hnRNP–RNA complex. Subsequently, they become involved in the processes of mRNA splicing, translation, transport, and biodegradation [105]. HnRNPs can be further subdivided into several subgroups, named in order from A to U, with relative molecular weights ranging from 34,000 kD to 120,000 kD [106,107]. HnRNPs have four unique RBDs: RRMs, quasi-RRM, the arginine glycine cluster (RGG box), and the nuclear protein KH structural domain [108,109,110]. Two highly similar RRMs form a βαββαβ structure in eukaryotic cells, and they contain two highly conserved shared RNP sequences [111]. Although these RRMs are highly similar, a significant difference exists in their affinity for RNA binding. In most cases, the RRMs preferentially bind RNA and can recognize longer motifs, whereas quasi-RRMs bind weakly and mainly assist in the binding of proteins to RNA. However, the disruption of RRM interactions or loss of either binding capacity can affect the function of hnRNPs [112]. The RGG region is the main auxiliary region of hnRNPs and mainly mediates the interaction of homologous or heterologous proteins with hnRNPs [110]. The KH domain forms the structure of βααββα, whose function is related to the splicing of target mRNAs [113]. In addition, several hnRNAs have a nucleoplasmic shuttle function, and they can form complexes with pre-mRNAs to assist mRNAs in the nucleoplasmic transport process (Figure 3) [114]. The M9 sequence is a special class of nucleoplasmic shuttle sequence that is distinct from the traditional NLS and is responsible for the bidirectional regulation of the transport of hnRNPs with shuttle function from the nucleus to the cytoplasm [115]. The nucleocytoplasmic shuttling function of hnRNPs relies on the complete M9 sequence. Single-amino-acid site mutations can disrupt normal protein input and output processes [116]. A part of the hnRNP also has several auxiliary sequences, such as Gly- and Pro-rich domains, which mediate protein–protein interactions, subcellular localization, and other functions [106]. Table 1 shows the structural and functional characteristics of the main members of the hnRNP family.

Numerous studies have confirmed that hnRNP family members are highly expressed in bladder cancer tissues or cell lines and involved in regulating the proliferation, invasion, metastasis, and apoptosis of bladder cancer. The high expression of hnRNP A3 was significantly associated with lymph node metastasis and poor prognosis in patients with bladder cancer [117]. Meanwhile, hnRNP F and hnRNP A2/B1 also have potential as prognostic markers for bladder cancer [118,119]. In addition, studies have shown that hnRNP U has the ability to deplete the sensitivity of bladder uroepithelial cancer cells to cisplatin, and inhibition of hnRNP U may be a potential treatment for cisplatin-resistant bladder cancer [120]. To describe the role of hnRNP family members in bladder cancer in detail, we will discuss and summarize the following aspects.

EMT is an important biological process for epithelial-derived malignant tumor cells to acquire migration and invasion abilities. This process mainly includes epithelial cell markers, such as the decreased expression of E-calmodulin (E-cadherin) and upregulated expression of mesenchymal cell-associated markers (vimentin and N-cadherin) [121]. In bladder cancer, this process is also regulated by hnRNPs. The overexpression of hnRNP L in bladder cancer cell lines will up-regulate the mesenchymal markers vimentin and snail, whereas the expressions of epithelial markers E-cadherin and β-catenin will be downregulated [122]. In bladder cancer tissues, the expression of hnRNP F is also significantly up-regulated, enhances stability, and promotes the translation of Snai1 mRNA, which promotes EMT and is significantly associated with poor prognosis among bladder cancer patients [123].

In addition to regulating EMT, hnRNP family members are involved in other invasion and metastasis mechanisms of bladder cancer. hnRNPA2/B1 can promote lymph node metastasis of bladder cancer in a VEGF-C-independent manner by stably binding to lncRNA-lymph node metastasis-associated transcript 2 and promoting its translation [124]. Moreover, hnRNP A1 promotes bladder cancer invasion by promoting the translation of hypoxia-inducible factor-1 through binding to target mRNAs [125].

Most hnRNP family members are overexpressed with enhanced proliferation of tumor cells, including bladder cancer cells. hnRNP F can promote bladder cancer cell proliferation and regulate the cell cycle by promoting the expression of targeting protein for xenopus kinesin-like protein 2 [126]. Furthermore, in bladder cancer, hnRNP F expression is regulated by the phosphatidylinositol-3 kinase/AKT pathway [127]. Moreover, hnRNP L inhibits apoptosis via suppression of caspase-3, -6, and -9 expression and enhances the mitogen-activated protein kinase signaling pathway, leading to proliferation and poor prognosis of bladder cancer [122]. Similarly, hnRNP K is highly expressed in bladder cancer and can promote bladder cancer proliferation and resist apoptosis by regulating the transcription and translation of mRNAs, such as cyclin D1 [128,129]. HnRNP K can also bind to the promoter of SOX2 mRNA and promote its translation, thereby promoting the proliferation and spheroid-forming ability of bladder cancer cells [130].

The increased number of reports on hnRNP family members in bladder cancer indicates a growing awareness of the important role of hnRNPs in the development of bladder cancer. HnRNPs influence bladder cancer cell proliferation, apoptosis, invasion, and metastasis through regulating the mRNA expression of different target genes. Given that members of the hnRNP family are commonly highly expressed in bladder cancer tissue, they can be utilized as a predictor of early-stage tumors. However, although hnRNP family members are expressed at different levels in various tissues, which is necessary for maintaining normal cell renewal [131], and are also highly expressed in stem cells, a number of issues still need to be clarified if they are to be used as therapeutic targets, such as targeted tumor tissue drug delivery, killing effect on normal tissue, and effects on embryos, fetuses, newborns, and children.

**Table 1 cancers-15-01150-t001:** The hnRNP family presented by their structural and functional characteristics, and its interacting lncRNAs.

hnRNP-	Molecular Weight (kDa)	RBD	Binding Sequence	Functions	References
A1	34	2 of RRM, RGG box, Gly-enrich domain	UAGGGA/U	Splicing Translational regulation Influence of mRNA stability	[111,116,118]
A2/B1	36/38	2 of RRM, RGG box, Gly-enrich domain	UUAGGG	Splicing Influence of mRNA stability	[124,125,129]
C1/C2	41/43	RRM	Poly U	Splicing Translational regulation Transcriptional regulation	[111,113]
D	44~48	2 of RRM	AU-rich domain	Translational regulation Telomere maintenance	[113]
E	39	3 of KH	Poly C	Splicing Translational regulation Transcriptional regulation Influence of mRNA stability	[119]
F	53	3 of quasi -RRM 2 of Gly-enrich domain	UUAGG	Splicing Telomere maintenance	[131,132]
K	62	3 of KH	Poly C	Splicing Translational regulation Transcriptional regulation Influence of mRNA stability	[117,133,134,135]
L	68	4 of RRM, Gly-enrich domain	CA-repeat sequence	Splicing Influence of mRNA stability	[127,136]
M	77	3 of RRM	Poly G/C	Splicing	[121]
R	31	3 of RRM, RGG box	UCUAUC	Translational regulation Transcriptional regulation	[113]
U	120	RGG, Gly-enrich domain	GGACUGCR-RUCGC	Splicing Transcriptional regulation	[115]

## 6. Others

Although four types of RBPs have been listed for their roles and functions in bladder cancer, more RBPs have been identified and reported in bladder cancer, including insulin-like growth factor messenger RNA binding protein 3 (IGF2BP3), nucleolin (NCL), and quaking (QKI). Moreover, it is reported that Fragile X-related gene 1 (FXR1) was identified as a novel cancer driver gene in urothelial carcinoma of the bladder (UCB) [132]; circ-SLC38A1 promotes BC cells invasion in vitro and lung metastasis in vivo in mice [133]. Although reports on the mechanism of action of these RBPs in bladder cancer are limited, we believe that they play a role in bladder cancer progression based on their role in other malignancies. Therefore, we also call for more basic and clinical research to focus on these RBPs.

### 6.1. IGF2BP3

IGF2BP3 is a member of the IGF2BP family, and its members all contain 2 RRM and 4 KH domains in their structure; these 4 KH domains are all binding GGC sequences [134]. Under normal or stress conditions, members of the IGF2BP family can act on target mRNAs in an N6-methyladenosine-dependent manner, promote their stability, and increase their intracellular content, thereby affecting gene expression [135,136]. However, currently, only IGF2BP3 has been reported in bladder cancer. The chromosomal localization of IGF2BP3 is 7p15.3.IGF2BP3, which is a typical multi-domain RNA-binding protein with specificity and diversity in recognizing targets and provides a good paradigm for the multivalent interactions of multi-domain RNA-binding proteins [137].

Compared with IGF2BP3, which has been widely reported in a variety of tumors such as breast, liver, and gastrointestinal tract tumors, pancreatic cancer, and lung cancer [138], knowledge on its role in bladder cancer is limited. Regardless, IGF2BP3 is highly expressed in bladder cancer patients and independently associated with bladder cancer recurrence, cancer-specific mortality, and all-cause mortality [139]. Knocking down the expression of IGF2BP3 in bladder cancer can increase apoptosis and induce cell cycle arrest, implying that it originally promoted cell proliferation via inhibiting apoptosis and regulating cell cycle [140].

### 6.2. NCL

NCL is one of the abundant proteins in the nucleolus, and it is widely distributed in the nucleolus, nucleoplasm, cytoplasm, and cell membrane of eukaryotic cells, participating in a variety of biological processes [141]. NCL is mainly distributed in the nucleus, is responsible for the transcription of rDNA, and has various roles in the biogenesis of ribosomes, including RNA polymerase I transcription, pre-rRNA processing, and ribosome assembly [142]. The human NCL gene is a haploid genome located on 2q12-qter, consisting of 14 exons and 13 introns [143]. The N-terminal domain of the NCL is majorly involved in DNA regulation and protein-to-protein reactions, and this region also contains a variety of highly phosphorylated sites that can be involved in cell cycle regulation [143]. The central domain holds four RRMs that bind to and regulate the transcription of specific mRNAs [144]. The C-terminal sequence and arrangement of NCL is not conserved, and its length is variable, usually interspersed with a large number of glycine, arginine, and phenylalanine residues; its main function is to help NCL interact during larger or more complex RNA localization [145]. NCL is highly expressed in a variety of tumors [146]; besides promoting tumor proliferation [147,148,149], it is an important anti-apoptotic protein that maintains tumor cell survival [143,150,151,152,153]. NCL has also been reported to be involved in various processes, such as angiogenesis [154,155,156], infiltration, and metastasis [157,158,159,160] of tumors.

In bladder cancer, increased expression of NCL boosts its aggressiveness and promotes its pulmonary metastasis [161], whereas blocking NCL expression in bladder cancer can inhibit bladder cancer proliferation and invasion [162,163,164]. However, the target mRNAs of NCL in bladder cancer, such as Rho factor [161,162] and/or MMP-2 [163], are also diverse. In addition, NCL has the potential to promote tumor proliferation by promoting the function of epidermal growth factor receptor [164,165]. The role played by NCL in bladder cancer is far less clearly explained than other tumors, and this area needs more research input. Regardless, the treatment targeting NCL has great potential in bladder cancer.

### 6.3. QKI

QKI protein is a subfamily of the signal transduction and activation of RNA (STAR) family [166]. The STAR protein family plays an important role in embryogenesis, tissue, and organ development [167,168]. The human Qki gene is localized on chromosome 6 and has nine exons, which can produce at least five transcripts by different splicing methods [169]. Three of the main transcripts are the most important, as they are named Qki-6, Qki-6, and QkiI-7 due to their sizes (5, 6, and 7 kb, respectively). The structure of QKI proteins is highly homologous to other members of the STAR family, with similar domains: an RRM (KH domain) flanked by QUA1 and QUA2 domains [169]. The QKI-5 protein is mainly localized in the nucleus, where it can bind to target mRNAs and retain them in the nucleus. This condition is possible because QKI-5 contains a nuclear localization signal peptide at its C-terminus, whereas QKI-6 and QKI-7, which lack this signal peptide, are mainly localized in the cytoplasm and are involved in target mRNA transport and regulate its stability [170]. The reason why QKI proteins can bind to target mRNAs is that these mRNAs have a sequence that can be specifically bound by QKI, namely 5′-A(C/A)UAA-3′; hence, this sequence is also called a quaking response element (QRE) [171]. Bioinformatic analysis showed that QKI can interact with more than 1000 mRNAs with QRE, and most of these are important molecules in cell-directed differentiation, proliferation, metastasis, and apoptosis [171,172]. QKI has a low expression in a variety of malignancies, and overexpression of QKI inhibits the proliferative, invasive, and migratory capacities of these tumors and promotes their apoptosis [173,174,175,176,177]. Thus, QKI may play an important role as an oncogenic factor in the development of malignant tumors.

In bladder cancer, the low expression of QKI is associated with advanced tumor TNM staging and poor overall survival, whereas its overexpression inhibits the ability of bladder cancer cells to grow and invade [178]. The oncogenic mechanism of miRNA-362-5p in bladder cancer is also related to QKI, which can promote the proliferative and invasive effects of bladder cancer via targeting binding and reducing QKI [179]. Cancer-associated fibroblasts (CAFs) are also an important component of the tumor microenvironment, secreting microfibrillar-associated protein 5 (MFAP5), a component of elastic microfibers, which is also an oncogenic factor in a variety of tumors [180], promoting tumor cell proliferation, invasion [181], and recruiting new blood vessels [182]. In bladder cancer, QKI can directly target binding to MFAP5 in CAFs and downregulate its expression, thus limiting tumorigenesis and progression [183]. The expression level of QKI in malignant tumors can be used as an important indicator to predict the occurrence and development of cancer, and it is expected to become a new target for tumor therapy, including that of bladder cancer. However, how to apply it effectively and precisely in clinical diagnosis and treatment needs further research. In particular, the functional differences between the subtypes of QKI in tumorigenesis development are rarely reported.

## 7. Discussion

In summary, a considerable number of RBPs have been revealed in bladder cancer. However, the complex regulatory network of these RBPs in bladder cancer remains to be fully elaborated. Moreover, numerous RBPs have been widely reported in other tumors without having been reported in bladder cancer. Examples include the Musashi protein family [184,185], Wilms Tumor 1 [186], and zinc finger protein homolog 36 family [187,188]. Although several RBPs have been reported in bladder cancer, a significant gap exists in the exploration of their mechanisms or the breadth of coverage of family members compared with their coverage in other tumors. Each of the three members of the IGF2BP family has been widely and intensively reported in acute leukemia [189], lung and esophageal cancer, and breast and gynecologic cancers [135]. However, in bladder cancer, as the only reported IGF2BP3, its molecular mechanism in regulating bladder cancer is still not elucidated. The emergence of new technologies promises to solve this problem, and CRIPR-Cas9 technology has been given high hopes to discover more RBPs. Several scholars have used this method to find new RBPs [190,191,192]. As the technology improves and becomes more widespread, CRISPR-Cas9 technology is no longer just a “finder” to discover more RBPs but a “detective” that is expected to reveal more of its functions. CRISPR-Cas9 revealed a novel function of RBP RBM39 as an RNA splicing factor [192].

Various RBPs promote bladder cancer progression by promoting bladder cancer proliferation, invasion, metastasis, and angiogenesis or activating EMT in bladder cancer, but there are still some RBPs that can inhibit clinical progression of bladder cancer by cell cycle blockade or modulating cell proliferation and invasion (Figure 4). The complex regulatory mechanism of RBPs makes research in this field challenging. This complex mechanism is multi-layered, and one RBP can have both multiple target mRNAs and multiple different modalities of regulation. The QKI protein functions mainly through the following four mechanisms: (1) synergizing [193,194] or competing [193,195] with miRNAs to exert regulatory effects; (2) the splicing factor QKI can regulate the expression of downstream targets via selective splicing and participate in several biological processes such as cell proliferation [196,197]; (3) inducing cell cycle arrest and inhibiting cell proliferation [166,173,198]; (4) mediating EMT and participating in the process of tumor cell proliferation and metastasis [199]. Whether these effects are features shared by QKI in all tumor cells or are influenced by tumor heterogeneity and whether they have different effects on various tumor cells remain unknown. Another key point is that numerous RBPs do not rely on classical RBDs for binding to target RNAs [200]. As a result, the bioinformatics prediction of RBP–RNA interactions is challenging [201]. As the field of RBPs in bladder cancer has been relatively slow to develop, the answers to these questions for bladder cancer are still unknown.

Many elements in oncology require study in terms of the their potential as a tumor marker or as a target for personalized therapy and other clinical issues. The study of RBPs is no exception. The potential of RBPs as biomarkers to assess patient prognosis or treatment response in different tumor contexts has been extensively explored [73,202,203,204,205]. Studies have also shown that several small-molecule inhibitors can selectively antagonize RBPs or RBP–RNA interaction in vitro, thereby exerting good anticancer effects [102,206,207]. These findings point to the same concept: RBPs have a very high potential in tumor prediction and personalized therapy. However, only with a deep understanding of their regulatory mechanisms, expression characteristics, and potential binding sites, among other issues, will it become possible to apply RBPs to clinical applications.

## 8. Conclusions

In conclusion, this article reviewed the effects and mechanisms of RBPs in bladder cancer. Several characteristics of RBPs that have been reported in bladder cancer are summarized. Thoughts on future research directions and problems faced in this field have been put forward. Hopefully, more researchers will be encouraged to devote attention to this field, and more bladder cancer patients will benefit from the development of RBP research.

## Figures and Tables

**Figure 1 cancers-15-01150-f001:**
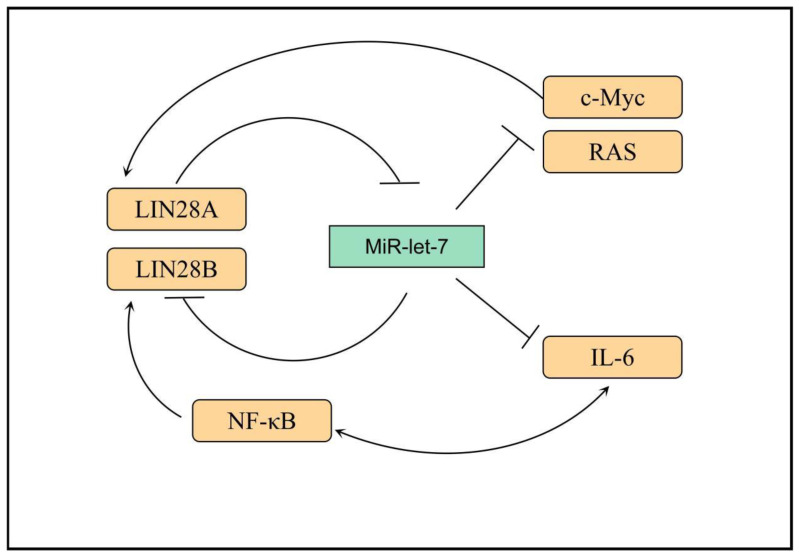
Transcriptional networks that regulate LIN28B expression. LIN28B expression is lost. In adult mammals, only a small subset of somatic cells exist where LIN28B expression occurs. Several transcription factors, such as MYC and NF-κB, promote LIN28B transcription, while REST and ESE3/EHF are transcriptional repressors. IL-6: interleukin-6, RAS: Resistance to audiogenic seizures, NF-κB: nuclear factor kappa-B, MYC: MYC protooncogene.

**Figure 2 cancers-15-01150-f002:**
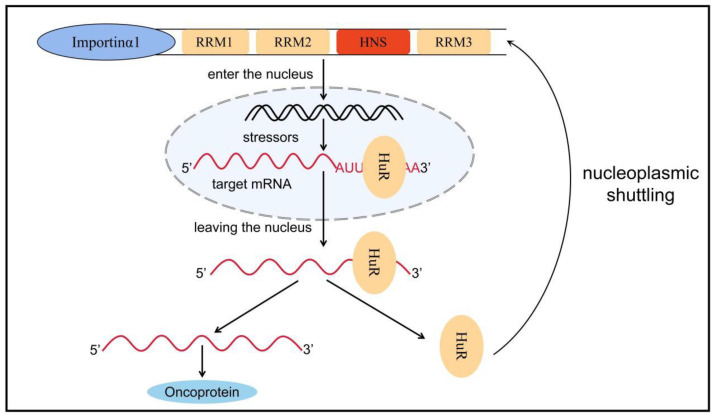
Nucleoplasmic transport of HuR. HuR is mainly distributed in the nucleus, but when stimulated by factors such as microenvironmental changes, HuR can bind to target mRNA to form a HuR-mRNA complex, which protects the target mRNA from degradation by nucleic acid exonucleases and transfers it from the nucleus to the cytoplasm via HNS. Subsequently, HuR dissociates from the target mRNA and rapidly returns to the nucleus with the assistance of transporter proteins such as importin α1. Nucleoplasmic translocation of HuR increases the stability of the target mRNA, promotes mRNA translation, and causes a variety of inflammatory phenotypes, ultimately promoting tumor formation and progression. RRM: RNA recognition motif, HuR: human antigen R, HNS: HuR nucleoplasmic shuttling sequence.

**Figure 3 cancers-15-01150-f003:**
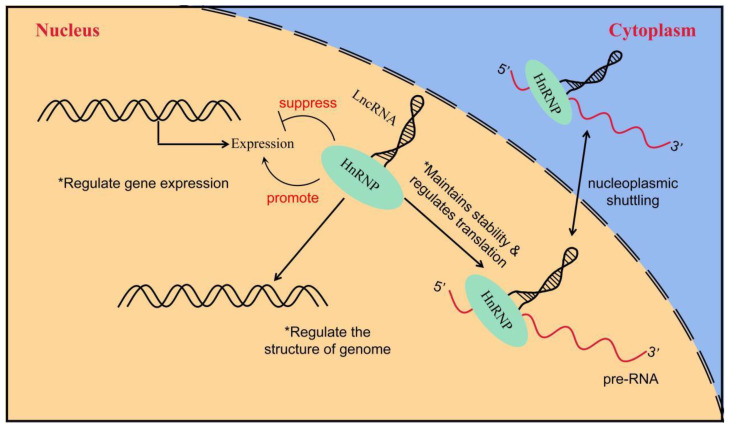
The way in which LncRNA interacts with hnRNP to regulate gene expression. LncRNA interacts with hnRNP to induce or inhibit gene expression; LncRNA regulates genome structure through interaction with hnRNP and indirectly affects gene expression; the interaction between lncRNA and hnRNP controls the stability and translation of mRNA.

**Figure 4 cancers-15-01150-f004:**
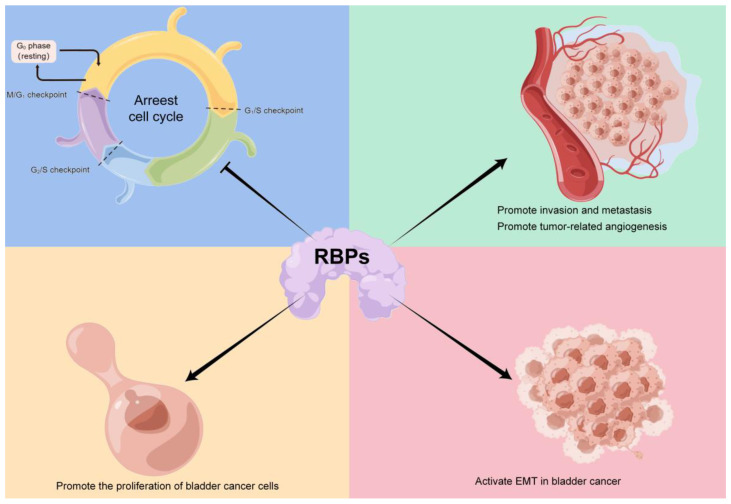
RBPs and bladder cancer. Various RBPs promote bladder cancer progression by promoting bladder cancer proliferation, e.g., HuR can stabilize HOTAIR mRNA; or promoting bladder cancer proliferation and inhibiting apoptosis, invasion, metastasis, e.g., LIN28A/LIN28B; or promoting angiogenesis, e.g., QKI promotes cancer-related fibroblasts to secrete MFAP5, the main component of elastic fibers, to recruit new blood vessels; or activating EMT in bladder cancer. There remain some RBPs that can inhibit clinical progression of bladder cancer by cell cycle blockage, e.g., RBM3 silenced can increase the number of G2/M stage cells and eventually lead to apoptosis or modulation of cell proliferation and invasion. This figure was made using *Figdraw*.

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
