# Peer review of "RNA-Binding Proteins in Bladder Cancer"

_cancers, 2023, doi:10.3390/cancers15041150_

Round 1
Reviewer 1 Report
RBPs are key regulators of transcription and translation in vital processes such as bladder cancer, they affect RNA life cycle leading to abnormal protein production. Understanding their regulatory mechanisms can lead to new therapies for cancer treatment. The article by Gao et al. reviews the effect of various RBPs on bladder cancer, summarising the types of RBPs involved in its progression and providing insights for further research.
Comments and suggestions are below:
1. The introduction section appears to be lengthy and verbose. Suggestions for simplification should be considered for improved clarity and conciseness, eg the first paragraph is unnecessary, the overly detailed discussion of RNA binding proteins can form a separate section.
2. A lot of citations from section 2-11 are not bladder cancer-related (considering your title is RBP in bladder cancer). The number of those should be kept at a minimum. It is suggested that the functional section of a particular protein can be combined with the one that discusses its association/roles in bladder cancer which could make the review more concise (eg. 2+3, 4+5, 6+7) and also consistent with section 8 and onwards.
In order to maintain consistency and concision within the review, a minimal number of citations from sections 2-11 that are not directly related to bladder cancer should be kept, in consideration of the title of the review being focused on RNA-binding proteins in bladder cancer. It may also be beneficial to combine the functional aspect of a specific protein with its association/roles in relation to bladder cancer (eg. sections 2+3, 4+5, and 6+7) which also aligns with the format of sections 8 and onwards.
Author Response
Response to the reviewers
Dear Editor and Reviewers:
Thank you very much for your valuable and insightful comments and suggestions which have greatly improved the quality of our manuscript. On behalf of all the authors, I would like to thank you for your efforts. We have carefully revised the manuscript according to your comments.
We now address both reviewers’ comments point counterpoint below.
Please download the Responding to review our answer.
To Reviewer 1
RBPs are key regulators of transcription and translation in vital processes such as bladder cancer, they affect RNA life cycle leading to abnormal protein production. Understanding their regulatory mechanisms can lead to new therapies for cancer treatment. The article by Gao et al. reviews the effect of various RBPs on bladder cancer, summarising the types of RBPs involved in its progression and providing insights for further research.
Comments and suggestions are below:
Question 1: The introduction section appears to be lengthy and verbose. Suggestions for simplification should be considered for improved clarity and conciseness, eg the first paragraph is unnecessary, the overly detailed discussion of RNA binding proteins can form a separate section.
Answer 1: Thanks very much for the comments. We have revised this part and we have added cited some references in the manuscript and highlighted them in yellow.
- Introduction
Tumor formation in humans is an extremely complex and multi-stage process that typically occurs over the years or decades. The histiocytes of normal individuals grad-ually develop into tumors with malignant phenotypes through evolution, through a process called tumor progression. Tumors can occur in various tissues of the human body, and the incidence increases with age. Only in very rare cases do cancerous cells progress to clinically visible tumor tissue with occupying lesions. Tumor progression is closely related to epigenetics, RNA post-transcriptional modification, protein post-translational modification, and other life processes. Tumors are not only regulated by these life processes but are also affected by normal biochemical reactions, reshaping cellular life activities and ultimately giving cells the ability to proliferate indefinitely.
Bladder cancer is one of the most common malignant tumors of the urinary sys-tem. The cancer statistics released in 2022 state that in the United States, bladder can-cer has an estimated incidence of 81 800 cases and a mortality rate of 17 100 cases [1]. These rates are slightly lower than those reported in 2021 [2]. In China, bladder cancer had an incidence of approximately 80 500 and a mortality rate of 32 900 in 2015 [3]. Bladder cancer is a broad concept that encompasses everything from low-risk non-muscle-invasive bladder cancer to high-risk primary invasive bladder cancer. Low- and intermediate-risk non-muscle invasive bladder cancer (NMIBC) patients face high re-currence rates, with 5-year event-free survival rates reaching 43% and 33% [4]. Metas-tasis of bladder cancer is a catastrophe that 50%–70% of muscle invasive bladder cancer (MIBC) patients have to face, and given the extremely high metastasis rate, the 5-year overall survival rate for advanced MIBC is 4.8% [5]. The research on bladder cancer not only requires the development of new biomarkers for diagnosis and molecular targets for personalized therapy but also in-depth studies of its progression, recurrence, me-tastasis, and other processes.
RBPs play crucial roles in the regulation of cellular life processes, especially RNA splicing, modification, transport, localization, stabilization, degradation, and transla-tion. Certain RBPs are expressed in a variety of cells to maintain essential cellular func-tions. Altered structure or disturbed expression of RBPs may cause different diseases, and this concept is reflected in tumorigenesis [6]. Given that RBPs can regulate post-transcriptional RNA, they can rapidly and efficiently alter gene expression in response to changes in the microenvironment. A single RBP can bind multiple targets, and dif-ferent combinations of several RNP interactions contribute to enhanced cellular recog-nition and responses to stress [7]. In addition, RBP can promote mRNA translation by recruiting specific translation signaling molecules [8]. By contrast, RBPs involved in the RNA-induced silencing complex can inhibit target mRNA translation while inducing its degradation [9, 10]. In several cases, two RBPs can compete in binding the same segment of mRNA, e.g., CUGBP2 and human antigen R (HuR) can bind to COX-2 mRNA, and in HCT-116 cell lines treated with radiotherapy, the RBP bound to COX-2 mRNA shifts mainly from HuR to CUGBP2 and inhibits the translation of COX-2 mRNA [11]. Certain RBPs play contrasting roles in a variety of different tumor cells, such as insulin like growth factor 2 mRNA binding protein 1, also known as IMP1, which promotes colorectal cancer [12] and liver cancer [13] occurrence, progression, and metastasis; however, it can inhibit the proliferation and metastasis of breast cancer [14]. Notably, a high stromal cell IMP1 expression in the colon cancer tumor microen-vironment suppresses tumorigenesis, whereas the deletion of stromal IMP1 forms a microenvironment that promotes colon carcinogenesis [15]. However, several RBPs are only expressed in specific tumor cells; PAT1 homolog 2 is highly expressed in renal chromophobe cancer, whereas dihydrouridine synthase 1 like is barely highly ex-pressed in bladder cancer [6].
Given that RBPs control gene expression mainly at the post-transcriptional level, the RRM domain can participate in blocking certain sites by binding RNA through a canonical RRM domain in its cap-binding complex [16, 17]. The RRM structural domain of RBP binds to RNA and alters the secondary structure of RNA; thus, RBP influences the entry of mRNA initiation factors into the ribosomal subunit [18], which ultimately regulates the activity of certain kinases in tumors. In addition, RRM plays a crucial role in precursor mRNA (pre-mRNA) splicing [19].
With RBPs playing an important role in biochemical processes, such as tumor-igenesis, progression, invasion, metastasis, and drug resistance, In particular, RBPs are involved in the development, progression and metastasis of bladder cancer[20], as well as in predicting survival of bladder cancer patients[21]; however, the role of RBPs in the development and progression of bladder cancer is unclear. We hereby summarize and review several RBPs that play a major role in the development and progression of bladder cancer. This review aims to provide a detailed characterization of the RBPs as-sociated with bladder cancer and focuses on their structure (especially RRM), function, interactions, causative pathogenic effects, and resulting treatment and prognosis. It al-so attempts to encourage clinicians and researchers to devote themselves in this field of study and provides a reference for researchers who aspire to pursue a career in this line of work.
Question 2: A lot of citations from section 2-11 are not bladder cancer-related (considering your title is RBP in bladder cancer). The number of those should be kept at a minimum. It is suggested that the functional section of a particular protein can be combined with the one that discusses its association/roles in bladder cancer which could make the review more concise (eg. 2+3, 4+5, 6+7) and also consistent with section 8 and onwards.
Answer 2: Thanks for your careful checks. In fact, these segmented paragraphs are a whole in our original manuscript. This error is due to paragraph segmentation during typesetting. We are sorry that these errors have caused misunderstanding. We have merged and revised these paragraphs. These revised paragraphs are as follows. Also, added references are highlight in yellow.
- RNA binding motif 3 (RBM3)
RBM3 is a glycine-rich cold shock protein whose expression can be stimulated by hypothermia, ischemia, or hypoxia [22-24]. RBM3 has two highly conserved RRMs, namely, RNP1 and RNP2, at the N terminus and an arginine-glycine-rich domain (RGG) at the C terminus [16]. The RGG structural domain mainly regulates the process of RNA cleavage and polyadenylate cyclization [25]. The RGG structural domain, espe-cially the part with arginine residues, is essential for mRNA export, because the ab-sence of a single arginine residue in the RGG structural domain can interrupt the shut-tling process of RBM3 between the nucleus and cytoplasm [25, 26]. RBM3 performs four main functions in tumors:
RBM3 can bind and affect the translation of mRNA. It influences mRNA stability and the translation of COX-2, interleukin (IL)-8, and vascular endothelial growth factor (VEGF) [27]. In general, RBM3 facilitates the translation of various mRNAs into pro-teins [26] [27, 28]. This promotion includes the following mechanisms: 1. binding to the 60S ribosomal subunit in an RNA-independent manner [28]; 2. increased formation of active polyribosomes [27]; 3. dephosphorylation of eukaryotic initiation factor (eIF2α); 4. promotion of eIF4E phosphorylation [28].
Under low-temperature conditions, RBM3 can alter miRNA levels and thus pro-mote protein translation. RBM3 binds to a precursor miRNA and facilitates its pro-cessing by the Dicer complex to form a mature double-stranded miRNA [29]. The regu-lation of miRNAs by RBM3 is two-sided: it can positively regulate most miRNAs, but reducing the level of RBM3 can promote the expression of a small number of tempera-ture-sensitive miRNAs, thereby preventing pathological hyperthermia [30]. These re-sults suggest that RBM3 is essential for the mitotic process of cells.
RBM3 can play a regulatory role in the cell cycle of G2/M transition. In colorectal cancer cells, RBM3 induces stem cell proliferation through a mechanism that increases β-catenin signaling by inhibiting glycogen synthase kinase-3 beta kinase activity [31]. By contrast, knockdown of RBM3 expression in the human HCT116 colon cancer cell line caused the increases in caspase-dependent apoptosis, nuclear cyclin B1 expression, and Cdc25c, Chk1, and Chk2 phosphorylation levels, which are a series of alterations suggesting that downregulation of RBM3 will prevent cell mitosis [27]. In vivo, embry-onic fibroblasts from RBM3-deficient mice showed a significant increase in the number of cells in the G2 phase [32]. These conclusions can also explain the higher sensitivity of tumors with a high RBM3 expression to chemotherapy than those with low or nega-tive expression [33].
When cells receive various external stimuli, unfolded proteins accumulate in the lumen of the endoplasmic reticulum and activate the unfolded protein response (UPR) to rescue cells. Sustained and/or intense endoplasmic reticulum stress (ERS) induces apoptosis [34]. The protein kinase R-like endoplasmic reticulum kinase (PERK)-eIF2α-C/EBP‐homologous protein (CHOP) signaling pathway plays an important role in UPR-induced apoptosis [35]. Under sustained ERS, RBM3 can inhibit the phosphoryla-tion of PERK and eIF2α, causing a decrease in the expression of CHOP and inhibiting UPR to avoid apoptosis [36]. This condition may be the reason why the UPR does not induce apoptosis despite its low-temperature activation; low-temperature-induced RBM3 may play an important role in this process [37]. Hypothermia can also alleviate ischemia-induced apoptosis by inhibiting the UPR [38].
In bladder cancer, the role of RBM3 is similar to that in other tumors. A clinical retrospective study including 259 bladder cancer patients showed that the low expres-sion of RBM3 was an independent factor for poor prognosis of bladder cancer [39]; this finding is closely related to the progression of bladder cancer and decreased overall survival of patients [40]. A similar study revealed that patients with a high expression of RBM3 were associated not only with a low tumor grade but also with a low risk of lymphovascular invasion (lymph node invasion) [[41]. The effect of RBM3 on bladder cancer may depend on several factors: 1. the expression level of RBM3 is closely related to tumor stage; 2. RBM3 silencing can increase the number of G2/M stage cells and eventually lead to apoptosis [33]; 3. RBM3 directly binds to a variety of mRNAs, thus regulating the activity of multiple kinases in tumors [42, 43]. For this stage, studies of RBM3 and bladder cancer have relied on the analysis of clinical samples in immuno-histochemistry, but more basic mechanistic studies are lacking. In particular, mecha-nistic studies on how RBM3 limits the development and progression of bladder cancer are limited. Clarifying these issues will not only further elucidate the mechanisms of bladder cancer development but also improve the current status of bladder cancer treatment and provide personalized therapeutic targets. In addition, the function of RBM3 in tumors and the mechanistic pathways it depends on can be elucidated and ex-tended to more tumor treatments.
- LIN28
LIN28 is a highly conserved RNA-binding protein in eukaryotes [44]. In a variety of mammals, including humans, LIN28 is divided into LIN28A and its homologous molecule, LIN28B [45]. Human LIN28A is encoded by the Lin28a gene, which is located on chromosome 1p36.11, and is mainly expressed in embryonic stem cells and embry-onic carcinoma cells [46, 47]. LIN28B is encoded by the Lin28b gene on chromosome 6q21 and is mainly expressed in the testis, placenta, and other tissues [45, 48]. LIN28A and LIN28B have highly similar protein structures: both have two functional domains, namely, a cold shock protein domain (CSD) and a retroviral zinc finger (cys-cys-his-cys, CCHC) domain [46]. After mutation of either domain, the other domain still has the function of binding RNA, suggesting that CSD and CCHC can participate in the RNA binding of LIN28 [49]. LIN28 protein is mainly localized in the cytoplasm [47]. However, it can also be present in RNPs, polyribosomes (polysome), P vesicles, and stress granules [50]. Meanwhile, LIN28B is mainly located in the nucleus, and it may exert its biological function through the cytoplasmic microprocessor [51]. However, the expressions of LIN28A and LIN28B are mutually exclusive, and tumor cells expressing LIN28A do not express LIN28B, and vice versa [51].
The miRNA let-7 family contains 12 miRNA members, which act as tumor sup-pressors and inhibit the expression of a variety of important oncogenes (including Ras, Myc, and so on) by binding to their 3′untranslated regions [52-54]. This function of let-7 is regulated by the RNA-binding protein LIN28 [55]. Overexpression of LIN28A or LIN28B is associated with a variety of tumors, leading to increased tumor aggressive-ness and poorer prognosis [56]. LIN28B has also attracted considerable attention as one of the downstream genes of nuclear factor (NF)-κB [57]. LIN28A and LIN28B can inhib-it the expression of oncogenes, such as Ras and Myc, by inhibiting let-7-miRNA [58, 59]. MiRNA let-7 acts as the main effector molecule of LIN28A and LIN28B, with which they form multiple feedback loops: 1. LIN28A/B can inhibit the maturation of let-7 through various mechanisms, whereas let-7 can inhibit the translation of LIN28A/B at the post-transcriptional level, reducing the protein expression level [60]; 2. after LIN28A/B inhibits the maturation of let-7, the inhibition of c-Myc by let-7 is relieved, and c-Myc can promote the transcription of LIN28A/B, forming a positive feedback loop [61, 62]; 3. NF-κB can induce the expression of LIN28B, and LIN28B inhibits the maturation of let-7, thus releasing the inhibitory effect of let-7 on the expression of IL-6, which can activate the expression of NF-κB, forming a positive feedback loop. Thus, linking inflammation and tumor further reveals the role played by inflammatory fac-tors in the malignant transformation of cells (Figure 1) [63].
In NMIBC, the inhibition of LIN28 significantly increases the expression level of let-7 and simultaneously inhibits the cell viability and migration ability of NMIBC [64]. Moreover, the expression of LIN28 in high-grade bladder transitional carcinoma (TCC) is significantly higher than that in normal bladder tissue and low-grade TCC; LIN28 can promote the progression and differentiation of NMIBC through the Lin28/let-7/c-Myc pathway [65]. LIN28B can also promote the expression of MYC through the LIN28B/let-7a pathway and promote the proliferation, invasion, and metastasis of bladder cancer cells [66]. LIN28A inhibits the expression of lysosome-associated mem-brane glycoprotein 1 in bladder cancer cells, thereby promoting bladder cancer prolif-eration, migration, and invasion [67]. In addition, LIN28A/LIN28B can promote bladder cancer proliferation, invasion, and metastasis by activating transforming growth fac-tor-β/Smad signaling pathway to drive epithelial–mesenchymal transition (EMT) [68]. LIN28B is also one of the downstream targets of macroH2A1, a histone variant; knock-ing down the expression of macroH2A1 can increase the expression of LIN28B. Knock-ing down the expression of macroH2A1 resulted in increased expression of LIN28B and enhanced tumorigenicity, radioresistance, degeneration of reactive oxygen species (ROS), and increased sphere formation ability of bladder cancer cells [69].
In conclusion, LIN28A and its paralog LIN28B are RBPs closely associated with tumors, and they exert their biological function by inhibiting the biosynthesis of members of the tumor suppressor gene let-7 miRNA family or changing the translation efficiency of the mRNA they bind. In bladder cancer, LIN28AB can promote tumor proliferation, metastasis, and invasion through a let-7-dependent mechanism and is resistant to radiotherapy and ROS. In other tumors, tumor cell lines highly expressing LIN28 exhibited resistance to chemotherapeutic drugs [70]. LIN28AB is a very poten-tial tumor therapy target. Inhibition of LIN28AB induced the regression of xenografted human tumors in mice [56]; however, future research needs to determine how LIN28AB/let-7 is precisely regulated.
- HuR
HuR is an embryonic lethal abnormal vision gene that includes four family mem-bers: HuB, HuC, HuD, and HuR. The first three are expressed mainly in neural tissues and germ cells and are associated with neurodevelopment, whereas HuR is commonly expressed in all human cells [71]. The human is located on chromosome 19p13.2, which is closely related to chromosomal translocations and tumor carcinogenesis in human tumors [72]. HuR contains three RRMs and a hinge region in which RRM1 and RRM2 bind to adenine- and uracil-rich elements (AU-rich elements, AREs) in the target mRNA. By contrast, RRM3 can bind to the polyadenylate tail of the target mRNA. In normal conditions, AREs can accelerate the poly-A tail of mRNA to undergo deadenyl-ation to destabilize mRNA [73]. Therefore, when HuR protein binds to these AREs, it can inhibit its own deadenylation and help mRNA to be protected from nuclease degradation during mRNA transport from nucleus to cytoplasm, thereby increasing mRNA stability and promoting mRNA translation; thus, HuR plays a role in post-transcriptional regulation [73, 74]. The hinge region between RRM2 and RRM3 contains a 52-amino acid HuR nucleoplasmic shuttling sequence, which is the main motif for post-translational modification of HuR and a key region for nucleoplasmic transport (Figure 2) [75, 76].
The nucleoplasmic transport function is essential for HuR to exercise its biological functions. Under physiological conditions, HuR is mainly distributed in the nucleus. When cells are damaged by radiation [77], depleted of nutrients or energy substances [78, 79], heat shock [80], viral infection [81], and chemotherapeutic drugs [82], or stimulated by cytokines [83], HuR shuttles from the nucleus to the cytoplasm. After activation through phosphorylation [84], methylation [85], and acetylation [86], HuR can shuttle from the nucleus to the cytoplasm, maintain the stability of bound mRNA, and promote mRNA translation [87, 88]. Given that HuR can interact with a variety of cy-tokines that promote tumor progression (such as survivin [89], COX-2 [90], VEGF [91], low-density lipoprotein receptor-related protein [92], etc.), mRNAs stably bind and promote its translation, which in turn leads to the abnormal distribution of tumor-promoting factors in the nucleocytoplasm of tumor cells. Therefore, inhibition of the nucleocytoplasmic shuttling of HuR may also become a potential tumor therapy.
In bladder cancer tissues, the nuclearly expressed HuR is not significantly corre-lated with any pathological features. However, the highly expressed HuR in the cyto-plasm plays an important role in the proliferation, progression, and survival of bladder cancer cells, and its expression is related to angiogenesis and clinical staging and grad-ing [93, 94]; in MIBC cells, the level of cytoplasmic HuR is significantly higher than that in NMIBC cells [95]. By contrast, the inhibition of HuR accumulation in the cytoplasm significantly increases the cytotoxicity of chemotherapeutic drugs, such as cisplatin and adriamycin, and inhibits the growth of xenografts in mouse bladder tumors [96]. In bladder cancer, HuR overexpression can promote bladder cancer proliferation, inva-sion, metastasis, and EMT by increasing the stability of polypyrimidine tract-binding protein 1 (PTBP1) to upregulate its expression [97], whereas destabilization of HuR causes a decrease in PTBP1 and inhibits bladder cancer progression [98]. In bladder cancer, HuR is an long non-coding whose other target RNA is thought to be HOX tran-script antisense RNA (HOTAIR). HuR can stabilize the stability of HOTAIR mRNA, promote bladder cancer proliferation, migration, and invasion, and inhibit apoptosis. Overexpression of HOTAIR can also increase the expression of HuR and promote its accumulation in the cytoplasm, thus enhancing HOTAIR expression and forming a positive HuR-HOTAIR feedback loop [99]. The role of HuR in the development of bladder cancer is largely dependent on the function of the mRNA it binds. The com-plex and extensive regulatory molecules in tumors endow HuR with a double-edged sword effect. HuR also enhances the stability and promotes the translation of ubiqui-tin-specific protease 8 by binding to its mRNA, which subsequently promotes the ubiquitination and degradation of SOX2 by acting as an E3 ligase, thereby inhibiting the invasive ability of bladder cancer [100]. HuR also upregulates miR-494 expression by stabilizing and promoting JunB mRNA translation. Up-regulated miR-494 destabi-lizes c-Myc mRNA and inhibits its translation, ultimately suppressing c-Myc-dependent matrix metalloproteinase (MMP)-2 expression, i.e., inhibiting bladder can-cer proliferation and invasion via the HuR/JunB/miR-494/c-Myc/MMP-2 axis [101].
The double-edged sword-like regulation mechanism of HuR on bladder cancer confirms the importance of HuR in bladder cancer. On the one hand, the activity of HuR is determined by its bound target mRNA, and on the other hand, it depends on the subcellular localization of HuR. More research is being focused on HuR as a target for tumor prevention and treatment. Nanocarriers [102] or cholesterol lipid nanocarri-ers [103] are used to transport siRNA to the tumor site, which allows it to bind to HuR mRNA and inhibit its translation, thereby inhibiting tumor growth. Inhibitors of HuR, such as MS-444 [104] or ChlA-F [100, 101], have also been tested for the treatment of tumors. Regardless of the type of targeted therapy strategy, further research is needed on its potential on- and off-target effects. However, at present, significant inhibitory effects have been achieved regarding the malignant phenotype of tumor cells caused by the abnormal expression of HuR. We look forward to more therapeutic strategies targeting HuR in bladder cancer.
- Heterogeneous nuclear RNPs (hnRNPs)
HnRNPs act as RBPs by binding to pre-mRNA to form the hnRNP–RNA complex. Subsequently, they become involved in the processes of mRNA splicing, translation, transport, and biodegradation [105]. HnRNPs can be further subdivided into several subgroups, named in order from A to U, with relative molecular weights ranging from 34,000 kD to 120,000 kD [106, 107]. HnRNPs have four unique RBDs: RRMs, quasi-RRM, the arginine glycine cluster (RGG box), and the nuclear protein KH structural domain [108-110]. Two highly similar RRMs form a βαββαβ structure in eukaryotic cells, and they contain two highly conserved shared RNP sequences [111]. Although these RRMs are highly similar, a significant difference exists in their affinity for RNA binding. In most cases, the RRMs preferentially bind RNA and can recognize longer motifs, whereas quasi-RRMs bind weakly and mainly assists in the binding of proteins to RNA. However, the disruption of RRM interactions or loss of either binding capacity can af-fect the function of hnRNPs [112]. The RGG region is the main auxiliary region of hnRNPs and mainly mediates the interaction of homologous or heterologous proteins with hnRNPs [110]. The KH domain forms the structure of βααββα, whose function is related to the splicing of target mRNAs [113]. In addition, several hnRNAs have a nu-cleoplasmic shuttle function, and they can form complexes with pre-mRNAs to assist mRNAs in the nucleoplasmic transport process (Figure 3) [114]. The M9 sequence is a special class of nucleoplasmic shuttle sequence that is distinct from the traditional NLS and is responsible for the bidirectional regulation of the transport of hnRNPs with shuttle function from the nucleus to the cytoplasm [115]. The nucleocytoplasmic shut-tling function of hnRNPs relies on the complete M9 sequence. Single-amino-acid site mutations can disrupt normal protein input and output processes [116]. A part of the hnRNP also has several auxiliary sequences, such as Gly- and Pro-rich domains, which mediate protein–protein interactions, subcellular localization, and other functions [106]. Table 1 shows the structural and functional characteristics of the main members of the hnRNP family.
Numerous studies have confirmed that hnRNP family members are highly ex-pressed in bladder cancer tissues or cell lines and involved in regulating the prolifera-tion, invasion, metastasis, and apoptosis of bladder cancer. The high expression of hnRNP A3 was significantly associated with lymph node metastasis and poor progno-sis in patients with bladder cancer [117]. Meanwhile, hnRNP F and hnRNP A2/B1 also have potential as prognostic markers for bladder cancer [118, 119]. In addition, studies have shown that hnRNP U has the ability to deplete the sensitivity of bladder uroepi-thelial cancer cells to cisplatin, and inhibition of hnRNP U may be a potential treat-ment for cisplatin-resistant bladder cancer[120]. To describe the role of hnRNP family members in bladder cancer in detail, we will discuss and summarize the following aspects.
EMT is an important biological process for epithelial-derived malignant tumor cells to acquire migration and invasion abilities. This process mainly includes epithelial cell markers, such as the decreased expression of E-calmodulin (E-cadherin) and up-regulated expression of mesenchymal cell-associated markers (vimentin and N-cadherin) [121]. In bladder cancer, this process is also regulated by hnRNPs. The over-expression of hnRNP L in bladder cancer cell lines will up-regulate the mesenchymal markers vimentin and snail, whereas the expressions of epithelial markers E-cadherin and β-catenin will be downregulated [122]. In bladder cancer tissues, the expression of hnRNP F is also significantly up-regulated, enhances stability, and promotes the trans-lation of Snai1 mRNA, which promotes EMT and is significantly associated with poor prognosis among bladder cancer patients [123].
In addition to regulating EMT, hnRNP family members are involved in other inva-sion and metastasis mechanisms of bladder cancer. hnRNPA2/B1 can promote lymph node metastasis of bladder cancer in a VEGF-C-independent manner by stably binding to lncRNA-lymph node metastasis-associated transcript 2 and promoting its transla-tion [124]. Moreover, hnRNP A1 promotes bladder cancer invasion by promoting the translation of hypoxia-inducible factor-1 through binding to target mRNAs [125].
Most hnRNP family members are overexpressed with enhanced proliferation of tumor cells, including bladder cancer cells. hnRNP F can promote bladder cancer cell proliferation and regulate the cell cycle by promoting the expression of targeting pro-tein for xenopus kinesin-like protein 2 [126]. Furthermore, in bladder cancer, hnRNP F expression is regulated by the phosphatidylinositol-3 kinase/AKT pathway [127]. Moreover, hnRNP L inhibits apoptosis via suppression of caspase-3, -6, and -9 expres-sion and enhances the mitogen-activated protein kinase signaling pathway, leading to proliferation and poor prognosis of bladder cancer [122]. Similarly, hnRNP K is highly expressed in bladder cancer and can promote bladder cancer proliferation and resist apoptosis by regulating the transcription and translation of mRNAs, such as cyclin D1 [128, 129]. HnRNP K can also bind to the promoter of SOX2 mRNA and promote its translation, thereby promoting the proliferation and spheroid-forming ability of blad-der cancer cells [130].
Increased number of reports on hnRNP family members in bladder cancer indi-cates a growing awareness of the important role of hnRNPs in the development of bladder cancer. HnRNPs influence bladder cancer cell proliferation, apoptosis, inva-sion, and metastasis through regulating the mRNA expression of different target genes. Given that members of the hnRNP family are commonly highly expressed in bladder cancer tissue, they can be utilized as a predictor of early-stage tumors. However, alt-hough hnRNP family members are expressed at different levels in various tissues, which is necessary for maintaining normal cell renewal [131], and are also highly ex-pressed in stem cells, a number of issues still need to be clarified if they are to be used as therapeutic targets, such as targeted tumor tissue drug delivery, killing effect on normal tissue, and effects on embryos, fetuses, newborns, and children.
- Others
Although four types of RBPs have been listed for their roles and functions in bladder cancer more RBPs have been identified and reported in bladder cancer, including insulin-like growth factor messenger RNA binding protein 3 (IGF2BP3), nucleolin (NCL), and quaking (QKI). Moreover, it is reported that Fragile X-related gene 1 (FXR1) was identified as a novel cancer driver gene in urothelial carcinoma of the bladder (UCB) [132]; circ-SLC38A1 promotes BC cells invasion in vitro and lung metastasis in vivo in mice [133]. Although reports on the mechanism of action of these RBPs in blad-der cancer are limited, we believe that they play a role in bladder cancer progression based on their role in other malignancies. Therefore, we also call for more basic and clinical research to focus on these RBPs.
Question 3: In order to maintain consistency and concision within the review, a minimal number of citations from sections 2-11 that are not directly related to bladder cancer should be kept, in consideration of the title of the review being focused on RNA-binding proteins in bladder cancer. It may also be beneficial to combine the functional aspect of a specific protein with its association/roles in relation to bladder cancer (eg. sections 2+3, 4+5, and 6+7) which also aligns with the format of sections 8 and onwards.
Answer 3: Similarly, these paragraphs are also divided due to typesetting. We have merged and modified this part and added some references. The added references are highlighted in yellow, and PMID is provided in this response. (For adjustment here, please refer to Answer 1-2.)
PMID: 35954405; PMID: 35194031; PMID: 36697384;PMID: 35130920;PMID: 33312357.
Once again, we appreciate your positive comments and valuable inputs. We hope the revised manuscript could meet your standard and be considered for final acceptance for publication in Cancers.

Reviewer 2 Report
The authors of the paper provided the detailed description of the roles and main mechanisms of the RNA-binding proteins involment in the cancer progression focused on the bladder cancer. To my mind it's an useful review summarazing published data for RBP in bladder cancer.
The additional point, which may be improve the paper:
Please, provide the general scheme of the regulation of bladder cancer progression by RBP, what processes are involved in this regulation?
And also please, describe the scheme in the Discussion section.
Author Response
Dear Editor and Reviewers:
Thank you very much for your valuable and insightful comments and suggestions which have greatly improved the quality of our manuscript. On behalf of all the authors, I would like to thank you for your efforts. We have carefully revised the manuscript according to your comments.
We now address both reviewers’ comments point counterpoint below.
Please download the Responding to review our answers.
To Reviewer 2
The authors of the paper provided the detailed description of the roles and main mechanisms of the RNA-binding proteins involment in the cancer progression focused on the bladder cancer. To my mind it's an useful review summarazing published data for RBP in bladder cancer.
The additional point, which may be improve the paper:
Question 1: Please, provide the general scheme of the regulation of bladder cancer progression by RBP, what processes are involved in this regulation?And also please, describe the scheme in the Discussion section.
Answer 1: Thanks for your suggestion. We have created a new Figure (Figure 4. in manuscript) to illustrate that RBPs can regulate the process of bladder cancer in many ways. At the same time, we also described the picture in the discussion. (The added description has been highlighted in yellow.)
“Various RBPs usually promote bladder cancer progression by promoting bladder cancer proliferation, invasion, metastasis and angiogenesis or activating EMT in bladder cancer, but there are still some RBPs that can inhibit clinical progression of bladder cancer by cell cycle blockade or modulating cell proliferation and invasion (Figure 4). The complex regulatory mechanism of RBPs makes research in this field challenging. This complex mechanism is multi-layered, and one RBP can have both multiple target mRNAs and multiple different modalities of regulation……”
(Figure was shown in the file of Responding)
Figure 4. RBPs and bladder cancer. Various RBPs usually promote bladder cancer progression by promoting bladder cancer proliferation, such as HuR can stabilize the stability of HOTAIR mRNA; or promoting bladder cancer proliferation and inhibit apoptosis, invasion, metastasis, such as LIN28A/LIN28B; or promoting angiogenesis, such as QKI promotes cancer-related fibroblasts to secrete MFAP5, the main component of elastic fibers to recruit new blood vessels; or activating EMT in bladder cancer. but there are still some RBPs that can inhibit clinical progression of bladder cancer by cell cycle blocked, such as RBM3 silenced can increase the number of G2/M stage cells and eventually lead to apoptosis, or modulating cell proliferation and invasion.

Reviewer 3 Report
In this article, Gao et al reviewed the role of RNA-binding proteins in bladder cancer. The authors make a detailed description of the different RNA-binding proteins that are relevant in bladder cancer. The figures are clear and descriptive. The text is well written, easy to read and understand.
Author Response
Dear Editor and Reviewers:
Thank you very much for your valuable and insightful comments and suggestions which have greatly improved the quality of our manuscript. On behalf of all the authors, I would like to thank you for your efforts. We have carefully revised the manuscript according to your comments.
o Reviewer 3
In this article, Gao et al reviewed the role of RNA-binding proteins in bladder cancer. The authors make a detailed description of the different RNA-binding proteins that are relevant in bladder cancer. The figures are clear and descriptive. The text is well written, easy to read and understand.
Answer 3: Thanks for your comments.
Once again, we appreciate your positive comments and valuable inputs. We hope the revised manuscript could meet your standard and be considered for final acceptance for publication in Cancers.

Round 2
Reviewer 1 Report
All previous feedback has been addressed to satisfaction.
It has come to my attention that Line 44-48 is borrowed from the team's publication Gao et al. Cancers 2022. Please paraphrase.
Author Response
Dear Editor and Reviewer:
Thank you very much for your valuable and insightful comments and suggestions which have greatly improved the quality of our manuscript. On behalf of all the authors, I would like to thank you for your efforts. We have carefully revised the manuscript according to your comments.
We now address both reviewers’ comments point counterpoint below:
To Reviewer 1
Question 1: It has come to my attention that Line 44-48 is borrowed from the team's publication Gao et al. Cancers 2022. Please paraphrase.
Answer 1: Thanks very much for the comments.We have revised the paragraph form Line 44 to 48 and updated the references. The revision as follow:
Bladder cancer is the commonest malignant tumor in urinary system. According to cancer statistics released in 2023, the estimated incidence of bladder cancer in the United States is 82,290 cases, and the mortality rate is 16,170 cases[1]. The incidence is higher than that in 2022, while the mortality is slightly lower than that in 2022[2]. According to the report published by China, the incidence of bladder cancer in 2020 was 91,893 cases, and the mortality was 42,973 cases[3].
Reference:
- Siegel R, Miller K, Wagle N, Jemal A: Cancer statistics, 2023. CA: a cancer journal for clinicians 2023, 73(1):17-48.
- Siegel R, Miller K, Fuchs H, Jemal A: Cancer statistics, 2022. CA: a cancer journal for clinicians 2022, 72(1):7-33.
- Xia C, Dong X, Li H, Cao M, Sun D, He S, Yang F, Yan X, Zhang S, Li Net al: Cancer statistics in China and United States, 2022: profiles, trends, and determinants. Chinese medical journal 2022, 135(5):584-590.
Once again, we appreciate your positive comments and valuable inputs. We hope the revised manuscript could meet your standard and be considered for final acceptance for publication in Cancers.
Sincerely
Yuanhui Gao; Shufang Zhang
2023-2-9
